# Signal inference in financial stock return correlations through phase-ordering kinetics in the quenched regime

Ixandra Achitouv[1], Vincent Lahoche[2], Dine Ousmane Samary[2,3]*

1 Institut des Systèmes Complexes ISC-PIF , CNRS, Paris, France, 2 Université Paris Saclay, Cea, Gif-sur-Yvette, France, 3 Faculté des Sciences et Techniques (ICMPA-UNESCO Chair) Université d'Abomey-Calavi, Bénin

* ousmanesamarydine@yahoo.fr

## Abstract

Financial stock return correlations have been analyzed through the lens of random matrix theory to differentiate the underlying signal from spurious correlations. The continuous spectrum of the eigenvalue distribution derived from the stock return correlation matrix typically aligns with a rescaled Marchenko-Pastur distribution, indicating no detectable signal. In this study, we introduce a stochastic field theory model to establish a detection threshold for signals present in the limit where the eigenvalues are within the continuous spectrum, which itself closely resembles that of a random matrix where standard methods such as principal component analysis fail to infer a signal. We then apply our method to Standard & Poor's 500 financial stocks' return correlations, detecting the presence of a signal in the largest eigenvalues within the continuous spectrum.

## 1 Introduction

Statistical field theory can be seen as a clever example of statistical inference [1], aiming to reproduce large-scale correlations in the collective behavior of a significant number of strongly interacting degrees of freedom. This perspective is motivated by information theory, with the famous example being the so-called $\phi_D^4$ field theory, which effectively describes the behavior of the $D$-dimensional Ising model near the ferromagnetic transition [2–4]. In this context, the effective field $\phi(x)$ represents the average of a large number of discrete spins at a scale where the precise interactions between them are completely blurred. More generally, equilibrium statistical physics can also be viewed as an instance of statistical inference, based on the maximum entropy distribution [5]. The core problem of statistical physics essentially involves extracting relevant features from a large set of particles that interact with each other. For example, the theories of ideal gases and the Navier-Stokes equations simplify the complexities of a gas or fluid composed of a vast number of particles into straightforward relationships between a small set of macroscopic parameters, such as pressure, temperature, density, or entropy. This overarching objective is analogous to data analysis, where the framework of large data sets presents a concrete example of a problem closely related to statistical physics. Standard methods for addressing this issue, such as principal component analysis (PCA), are

**Funding:** The author(s) received no specific funding for this work.

**Competing interests:** The authors have declared that no competing interests exist.

specifically designed to extract the degrees of freedom that dominate the correlation spectrum of a data set (see, for instance, [6,7] and references therein). However, PCA requires a clear separation between the "relevant" degrees of freedom and those that can be disregarded (the "noise"). This condition fails in cases of nearly continuous spectra, where PCA cannot establish a clear boundary between the degrees of freedom see [8,9] and Fig 1. In this figure, we present two typical empirical spectra encountered in data analysis. On the left, the signal is distinct from the bulk, allowing standard PCA to isolate it effectively. On the right, however, the continuity of the spectra renders PCA ineffective at distinguishing between degrees of freedom. Here, "continuity" means that the spacing between eigenvalues within a connected component is typically of order $1/N$, where $N$ is the size of the empirical correlation matrix (ECM), which is *positive definite*. Let us now discuss the general questions arise in the signal detection apply to financial stock. Signal detection in financial markets refers to the process of identifying meaningful patterns or anomalies in market data that can inform trading decisions. These signals often originate from price movements, volume fluctuations, volatility, news sentiment, or macroeconomic indicators. The goal is to distinguish between "true" signals that indicate future price action and "noise" that may result from random fluctuations. Signal detection theory, originally developed in psychophysics, has been adapted to financial modeling to address uncertainty in noisy environments [10]. In finance, this theory is used to model the trade-off between detecting a real trading opportunity (true positive) and avoiding false signals (false positives). One of the most common signal detection methods in financial markets is technical analysis, which includes indicators such as Moving Averages, Relative Strength Index, Bollinger Bands [11]. These indicators help traders spot trends, momentum changes, or overbought/oversold conditions. Machine learning models, including Support Vector Machines, Random Forests, and deep learning, have increasingly been used to detect complex nonlinear patterns in high-dimensional financial data [12]. Statistical models, such as Autoregressive Integrated Moving Average, Generalized Autoregressive Conditional Heteroskedasticity, and Kalman filters, are employed to extract signals from time series data by modeling dependencies and volatility [13,14]. Natural Language Processing techniques are used to detect sentiment from financial news or social media, providing early signals about market direction [15]. These textual signals are often used in combination with

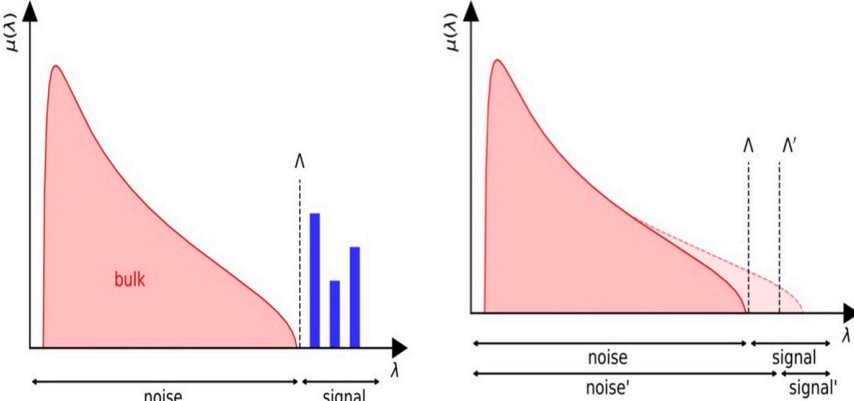

**Fig 1. Empirical spectra can exhibit some localized spikes (left) out of the continius spectrum (i.e. bulk noise, in red) made of delocalized eigenvectors (i.e. relevant information, in blue), in which case the cut-off Λ provides a clean separation between delocalized eigenvectors and localized ones.** For nearly continuous spectra (right), the position of the cut-off Λ is more difficult to understand.

quantitative models for improved accuracy. Key challenges in signal detection include the risk of overfitting, data snooping bias, and non-stationarity in financial time series.

Recently, a series of papers have approached the problem of spectral tails through statistical inference, based on an unconventional local Euclidean field theory [9,16–20]. In [9], the authors proposed to use an effective field theory model capable of addressing the full spectrum. This model resembles an equilibrium non-local field theory, characterized by a specific $O(N)$ invariance. In this article, we considers a non-equilibrium version of this field theory and we explore the link between the presence of a signal and the ability of the out-of-equilibrium field to reach equilibrium over a long time. For this analysis, we focus on financial data, specifically the correlation of returns in the Standard & Poor's (S&P) 500 index. Indeed, it has been shown that the distribution of the eigenvalues is approximately captured by a random matrix spectrum, except for a few spikes corresponding to the "market" mode or collective modes (e.g., [21–23]).

The manuscript is organized as follows: In Sect 2, we define the model inspired by [24] and provide the theoretical frameworks used throughout the paper. In Sect 3, we construct the formal solution of the Langevin equation in the quenched regime for a real vector of size $N$, where disorder is represented by a Wigner matrix. We specifically investigate the self-consistent evolution equation for the effective potential arising from the self-averaging of the square length. In Sect 4, we apply the previous formalism to study financial markets, focusing on the S&P 500 while varying the signal-to-noise ratio by perturbing correlations with an appropriate Brownian motion. Finally, conclusions and perspectives are summarized in Sect 5. Appendices A and B provide additional material. The data and codes used in this work can be accessed at https://github.com/Eleo22/RN-Finance.

## 2 The model

The essence of this section is based on the work presented in [9], where we constructed a maximum entropy estimate (the least structured one) for the empirical probability distribution of the microscopic degrees of freedom underlying a given ECM spectrum. This inferred distribution resembles a discrete version of the standard Ising model and is described by the partition function:

$$Z[J] := \int_{-\infty}^{+\infty} \prod_{i=1}^{N} d\phi_i \, e^{-S[\phi] + \sum_i J_i \phi_i}, \tag{1}$$

where:

$$S[\phi] = \frac{1}{2} \sum_{i,j} \phi_i \tilde{C}_{ij}^{-1} \phi_j + g \sum_i \phi_i^4 + \mathcal{O}(\phi_i^6). \tag{2}$$

The bare kinetic operator $\tilde{C}^{-1}$ is such that the (nonperturbative) inference condition holds:

$$\langle \phi_i \phi_j \rangle_c \equiv C_{ij}, \tag{3}$$

where $C_{ij}$ is the $(i,j)$ entry of the ECM. At the zero order in the perturbation theory, we have $C_{ij} = \tilde{C}_{ij}$, but quantum corrections arises and broke this property to higher orders. The formal relation between them is given by the so-called Dyson equation:

$$C^{-1} = \tilde{C}^{-1} - \Sigma, \tag{4}$$

where the matrix $\Sigma = \mathcal{O}(g)$ is the *self energy*. We assume that $C$ is closed enough to a positive Gaussian noise, or more precisely that eigenvectors are non-localized enough to remains close

to the Marchenko-Pastur (MP) law (see Appendix A). In the perturbative regime, it is suitable to assume that $\tilde{C}$ inherits from these properties. Then, denoting by $u^{(\mu)}$ the eigenvector of $\tilde{C}^{-1}$ such that:

$$\sum_{j=1}^{N} \tilde{C}_{ij}^{-1} u_j^{(\mu)} = \lambda_\mu u_i^{(\mu)}, \tag{5}$$

where $\lambda_\mu \geq 0$. We expect that the distribution for components $u_i^{(\mu)}$ is close enough to the Porter-Thomas distribution, see Appendix A. It is suitable to work in the eigenspace rather than in the "real space", and we introduce the field $\varphi_\mu := \sum_i \phi_i u_i^{(\mu)}$ such that the classical action $S$ becomes:

$$S[\varphi] = \frac{1}{2} \sum_{\mu=1}^{N} \lambda_\mu \varphi_\mu^2 + g \sum_{\{\mu_i\}} V_{\mu_1 \mu_2 \mu_3 \mu_4} \prod_{k=1}^{4} \varphi_{\mu_k} + \mathcal{O}(\varphi^6), \tag{6}$$

where the overlap tensor $V_{\mu_1 \mu_2 \mu_3 \mu_4}$ is:

$$V_{\mu_1 \mu_2 \mu_3 \mu_4} := \sum_{i=1}^{N} u_i^{(\mu_1)} u_i^{(\mu_2)} u_i^{(\mu_3)} u_i^{(\mu_4)}. \tag{7}$$

Because of the non-localized structure of the eigenvecteurs, the relevant values for $V_{\mu_1 \mu_2 \mu_3 \mu_4}$ are for $\mu_1 = \mu_2 = \mu_3 = \mu_4$, when the indices are equal in pairs, as can be easily checked numerically, for instance, with a Gaussian random matrix [9]. Furthermore, the case where all the indices are equal provides sub-leading quantum corrections and can be removed without any ambiguity (They cannot be perturbatively generated from the second kind of combinations in the large $N$ limit; see [9] again.). Hence, we can keep only configurations where the indices are equal in pairs

$$S[\varphi] \approx \frac{1}{2} \sum_{\mu=1}^{N} \lambda_\mu \varphi_\mu^2 + 3g (\varphi \cdot \varphi)^2 + \mathcal{O}(\varphi^6), \tag{8}$$

where $\varphi \cdot \varphi := \sum_\mu \varphi_\mu^2$, is the length square of the field. This observation generalizes for higher interactions which express in terms of $O(N)$ invariants. Note that in this approximation, it is shown that the self energy is the solution of a closed equation in the large $N$ limit, which can be solved exactly. The solution is diagonal (i.e. independent from the "momentum" $\lambda_\mu$ in the diagonal basis). Then, in this limit, $u_i^{(\mu)}$ are also eigenvectors for $C^{-1}$, and quantum corrections are all contained in the *effective mass* – see [9] for more details, and the above property is indeed a well known property of $O(N)$ models [25]. Denoting as $x_\mu$ the eigenvalues for $C$, and by $x_+$ and $x_-$ respectively largest and smallest eigenvalues nearly continuous component of its spectra, $\lambda_\mu$ is defined as:

$$\lambda_\mu := (x_\mu - x_-)^{-1} - (x_+ - x_-)^{-1}. \tag{9}$$

In this work, in line with the analysis presented in [20], we intend to adopt a different perspective, questioning the relationship between the presence of a signal and the stability of the maximum entropy distribution, rather than conjecturing it. To this end, and following [20], we will consider an out-of-equilibrium process described by a Langevin-like equation that models the motion of a classical particle in an N-dimensional random energy landscape.

Formally, in the eigenspace, this equation reads:

$$\frac{dq_\mu}{dt} = -[\lambda_\mu + \ell(t)]q_\mu(t) + \eta_\mu(t),$$

(10)

where $q_\mu := \sum_{i=1}^{N} q_i u_i^{(\mu)}$ is the projection along the eigenvector $u_i^{(\mu)}$ and $\eta_\mu(t)$ is a Gaussian white noise with zero means and 2-point correlation function:

$$\overline{\eta_\mu(t)\eta_\nu(t')} = 2T\delta_{\mu\nu}\delta(t-t').$$

(11)

Such a kind of equation is considered for instance for $p$-spin models [26,27], with the difference that $\lambda_\mu$ is here the inverse eigenvalues of $\tilde{C}$. Hence, choosing for $\ell(t)$,

$$\ell(t) = \sum_{n=0}^{n_{\max}} h_n a^n(t); \quad a(t) := \frac{1}{N}\sum_{\mu=1}^{N} q_\mu^2(t),$$

(12)

the equilibrium probability distribution for $q$ is [3]:

$$\rho[q] = e^{-\frac{1}{2}\sum_\mu \lambda_\mu q_\mu^2 - \sum_n \frac{h_n}{N^{n-1}} a^{n+1}}.$$

(13)

This distribution matches the $O(N)$ equilibrium theory considered above. Note that it is reasonable to assume that the spectrum for $\tilde{C}^{-1}$ start to 0, corresponding to the "mass" $h_0$. Thus, $\lambda \in [0, +\infty[$, and we denote the corresponding empirical distribution by $\rho(\lambda)$. Furthermore, we assume that the equilibrium statement regarding the large $N$ limit holds and that $\rho(\lambda)$ is also given by the (suitably shifted) spectrum of the ECM.

In the next section, we investigate analytical solutions of this Langevin like equation using random matrix theory, and especially statement about large $N$ Wishart matrices.

## 3 Analysis in the quenched regime

In this section, we construct the analytic solution of equation (10) by following the general method outlined in [24], drawing inspiration from Bray's solution for phase-ordering kinetics [28] and the Cugliandolo-Dean solution for the $p = 2$ spherical spin glass [26]. The central assumption for deriving this solution is the self-averaging property of $a(t)$, which decouples from $q_\mu(t)$ in the equation of motion (10) (quenched regime). We expect this hypothesis to be sufficiently realistic in the continuous limit, as $N \to \infty$, which is the limit of primary interest to us. The formal solution is then given, assuming once again $\lambda > 0$:

$$q_\mu(t) = q_\mu(0)\,e^{-\lambda_\mu t}e^{-g(t)} + \int_0^t dt'\, e^{-\lambda(t-t')}\,e^{g(t)-g(t')}\eta_\mu(t'),$$

(14)

where:

$$g(t) = \int_0^t dt'\,\ell(t').$$

(15)

Note that for large $N$, eigenvalues $\lambda$ are assumed to display accordingly with the Wigner semi-circle law with variance $\sigma^2$ i.e.:

$$\frac{1}{N}\sum_\lambda f(\lambda) \to \int_{-2\sigma}^{2\sigma} \mu(\lambda)f(\lambda)d\lambda, \quad \mu(\lambda) = \frac{\sqrt{4\sigma^2 - \lambda^2}}{2\pi\sigma^2}. \tag{16}$$

Also, $q_\mu(t)$ depends only on the numerical value $\lambda_\mu$ of the corresponding eigenvalue i.e. $q_\mu(t) \equiv q(\lambda_\mu, t)$. In the continuous regime, the square averaging $a(t)$ can be expressed in terms of the empirical distribution $\mu(\lambda)$:

$$a(t) = \int_0^\infty d\lambda\, \rho(\lambda)\langle q^2(\lambda, t)\rangle. \tag{17}$$

Using the solution (14) for $q(\lambda, t)$, the previous equation leads to a formally closed equation for $a(t)$:

$$\boxed{a(t) = G^{-1}(t)\left[H(t) + 2TF(t)\right]} \tag{18}$$

where $G(t) := e^{2g(t)}$,

$$H(t) := \int_0^\infty \rho(\lambda)e^{-2\lambda t}d\lambda, \tag{19}$$

and where $F(t)$ looks as the convolution product of the above function $G(t)$ and $H(t)$ namely:

$$F(t) = \int_0^t dt'\, H(t - t')G(t'). \tag{20}$$

**Remark.** *In the general case, the spectrum of the correlation matrix exhibits a bulk, quasi-continuous and a series of connected components (sometimes reduced to a single eigenvalue, as is the case for the largest eigenvalue). Our study focuses on the bulk, i.e. the continuous part of the spectrum. We will therefore impose a cut-off for a certain eigenvalue $\Lambda_c$ at the level of the correlation spectrum, thus retaining only a number $1 \ll N_c < N$ of eigenvalues. The previous formulas thus become in practice:*

$$H(t) := \frac{1}{N_c}\sum_{\mu=0}^{N_c} e^{-2\lambda_\mu t}. \tag{21}$$

For a purely uncorrelated distribution well described by the Marchenko-Pastur (MP) distribution (see Appendix A), the function $H(t)$ can be computed exactly. For variance $\sigma^2$ and setting $q \equiv N/P = 1$, we find:

$$H(t) = \frac{\sqrt{\frac{2}{\pi t}}\sigma - e^{\frac{t}{2\sigma^2}}\operatorname{erfc}\left(\frac{\sqrt{t}}{\sqrt{2}\sigma}\right)}{4\sigma^3}. \tag{22}$$

Accordingly with [24], one obtain a more tractable equation from the observation that long time physics must be dominated by configurations such that:

$$\ell(t) = h_0 + h_1 a(t) = 0. \tag{23}$$

This simply means that trajectories must be trapped by the minimum of the potential, and for $h_0 < 0$, it is for the non-vanishing value : $a(t) \equiv a_0 := -h_0/h_1$, and from (18), we get:

$$G(t) \approx \frac{1}{a_0} H(t) + \frac{2T}{a_0} F(t) \,. \tag{24}$$

This equation can be formally solved using the Laplace transform, which is defined for sufficiently well-behaved functions $f(t)$ as:

$$\bar{f}(p) := \int_0^\infty dt \, e^{-pt} f(t) \,. \tag{25}$$

We arrive to:

$$\bar{G}(p) = \frac{1/2}{\frac{1}{2} a_0 \bar{H}^{-1}(p) - T} \,. \tag{26}$$

Where $\bar{H}(p)$ can be analytically computed for MP law, and is a decreasing function with respect to $p$. For $q = 1$, we get:

$$\bar{H}(p) := \frac{1}{2\sqrt{2}\sqrt{p}\sigma^2 + 2\sigma} \,. \tag{27}$$

Then, $\bar{H}^{-1}(p)$ is minimal for $p = 0$, and because $\bar{G}(p)$ is positive definite, we must have (As pointed out in [24], the estimation is pessimistic):

$$T < T_c := \frac{1}{2} a_0 \bar{H}^{-1}(0) \,, \tag{28}$$

and we get $T_c = a_0 \sigma$ for MP. Furthermore, $\bar{H}(p) - \bar{H}(0) \sim \sqrt{p}$ and $\bar{G}(p) - \bar{G}(0) \sim \sqrt{p}$. Hence, because of the standard results about the asymptotic expression of inverse Laplace transforms near the origin (see B), we get $G(t) \sim t^{-3/2}$. Furthermore, the late time 2-points correlation $K(t)$ defined as:

$$K(t) := \int_0^\infty d\lambda \, \rho(\lambda) \langle q(\lambda, t) q(\lambda, 0) \rangle \,, \tag{29}$$

behaves as $K(t) \sim t^{-3/4}$ for MP below the critical temperature. This means that the memory of the initial condition is long i.e. it has infinite exponential time life. Finally, let us notice that the method break-down in the high temperature regime, and the origin of this failure can be traced from the behavior of $G(t)$, which in that regime diverges exponentially. In other words, above $T_c$ the system is expected to relax toward equilibrium accordingly with an exponential law, and the memory of the initial condition has a finite time life.

Let us remark that the same result maybe obtained by the statement that the correlator $G(t)$ admits the low temperature expansion

$$G(t) = \sum_{n=0}^{\infty} T^n G^{(n)}(t) \,, \tag{30}$$

assumed to have a finite radius of convergence, which we identify with the critical temperature below. The functions $G^{(n)}(t)$ can be constructed recursively from the closed equation (18). It is straightforward to check that functions $\{G^{(n)}(t)\}$ satisfy the following recurrence relations:

$$G^{(0)}(t) = -\frac{h_1}{h_0}H(t), \quad G^{(n)}(t) = -\frac{2h_1}{h_0}\int_0^t dt' H(t-t')G^{(n-1)}(t'),\, n > 0\,. \tag{31}$$

Hence, the Laplace transform $\bar{G}(p)$ of $G(t)$ reads:

$$\bar{G}(p) \quad = -\frac{h_1}{h_0}\bar{H}(p)\sum_{n=0}^{\infty}\Big(-\frac{2h_1}{h_0}T\bar{H}(p)\Big)^n = -\frac{h_1}{h_0}\frac{\bar{H}(p)}{1 + \frac{2h_1}{h_0}T\bar{H}(p)}\,. \tag{32}$$

Because $\bar{H}(p)$ is a decreasing function of $p$, the radius of convergence $R$ is fixed by setting $p = 0$, and the series formally resumes as for $T < R \equiv T_c$.

## 4 Signal detection threshold for financial markets

Before examining how this topic relates to our context, let us first briefly review signal detection as described in reference [9]. PCA works well when the covariance matrix spectrum has a few dominant eigenvalues ("spikes") that clearly separate signal from noise. This is evident when a small number of eigenvalues capture most of the variance, seen as a gap in the cumulative fraction. In many real datasets, however, the spectrum is nearly continuous, there are many relevant features spread over a wide range of eigenvalues without clear gaps. This makes it hard for PCA to cleanly separate signal from noise, as noisy and relevant directions mix strongly. This difficulty is similar to what happens in critical phenomena in statistical mechanics, where scales do not decouple cleanly. RG is a powerful method for addressing this by coarse-graining degrees of freedom hierarchically. Standard random matrix-based noise models (e.g., Marčenko–Pastur law) help, but have limitations: They need detailed noise modeling, they must handle data sparsity, they can't fully separate strongly mixed relevant and noisy directions in nearly continuous spectra.

We then uses scalar field theory as an analogy. There, RG flow shows how interactions become relevant or irrelevant based on the energy spectrum's shape and the space's dimension [8]. Similarly, for data with a nearly continuous spectrum, one could think of describing correlations through an effective field theory that captures interactions among degrees of freedom. Instead of focusing on identifying isolated signals in the spectrum, the idea is to model the full structure using a field theory framework, treating the density of eigenvalues as analogous to a particle energy spectrum. This provides a universal description of interactions in the data, potentially overcoming PCA's limitations for continuous spectra. Note that this idea connects to previous work where local interactions in momentum space were modeled. Reference [9] extends the approach by considering non-local effects and includes more numerical results and applications. As result, it is possible to understand signal detection by the significant changes on the universal properties of noise models, in particular for the number of relevant couplings by which asymptotic states in the IR are distinguished. That is reminiscent of the physics of critical phenomena, and makes it possible to consider signal detection as a phase transition, breaking the native $\mathbb{Z}_2$ symmetry of models based on a principle of maximum entropy. Moreover, the RG allows a natural understanding of the existence of a detection threshold due to the existence of a compact subset of physically acceptable initial conditions, included in the symmetric phase.

In the following analysis, we consider stocks from the S&P 500 index over the period from January 1, 2019, to January 1, 2024, using data downloaded from Yahoo Finance. We exclude stocks that were not present for the entire time range, leaving us with 485 stocks and 1,258 days of closing prices for each stock.

Instead of examining the system's behavior under a single spectrum, we aim to compare different regimes corresponding to varying signal-to-noise ratios. Our general approach will involve constructing an interpolation between two extreme regimes: one that is highly correlated and another that is completely uncorrelated (spurious noise, which can be well-described by random matrix theory).

## 4.1 A model to vary the signal-to-noise ratio in the correlation of stock returns

We consider the Geometric Brownian Motion (GBM) model, which is a widely used stochastic process in finance for modeling asset prices. It assumes that stock prices follow a log-normal distribution. In its integral form, GBM describes the evolution of stock prices over time and is given by:

$$S_t^{\text{GBM}} = S_0 \exp\left(\left(\mu - \frac{\sigma^2}{2}\right)t + \sigma W_t\right), \tag{33}$$

where $S_0$ is the initial stock price, $\mu$ is the drift coefficient representing the expected return of the stock, $\sigma$ is the volatility coefficient representing the standard deviation of the stock's returns, and $dW_t$ is a Wiener process (or Brownian motion) (e.g., [29]), representing the random component of stock price changes.

In this model, there is no correlation between the generated stock prices, which is not the case in actual financial stocks. For each S&P 500 stock, we construct a simulated walk with varying degrees of 'noise' or 'temperature' as:

$$S_t^{\text{sim}} = \beta S_t^{\text{GBM}} + (1 - \beta)S_t \,\forall \beta \geq 0, \tag{34}$$

where $S_t$ is the actual stock price. The GBM stock price is computed using $S_0$, $\mu$, and $\sigma$, which are computed from the historical data. In contrast, we generate fully correlated walks by using the same random seed in equation 33, which we refer to as $S_t^{\text{GBM corr}}$, and apply the same weighting scheme:

$$S_t^{\text{sim}} = -\beta S_t^{\text{GBM corr}} + (1 + \beta)S_t \,\forall \beta < 0. \tag{35}$$

In the left panel of Fig 2, we show the correlation matrix of the stock price log-returns:

$$C_{i,j} = \frac{< r_i(t)r_j(t) > - < r_i(t) >< r_j(t) >}{\sigma_i \sigma_j} \tag{36}$$

where $r_i(t) = \log[S_i(t)] - \log[S_i(t-1)]$, and $\sigma_i$ is the standard deviation of the stock i, computed over the period under consideration. The different figures represent various values of $\beta = [1, 0.6, 0, -1]$. The right panels show the distribution of the eigenvalues for these correlation matrices, where we display two fits: the Marčenko-Pastur (MP) distribution and the "rescaled" MP distribution, which subtracts the contribution of the larger eigenvalues [21]. We also mark the position of the cut-off value $\Lambda_c$ (dashed vertical red line), which defines the threshold of the continuous spectrum (eigenvalues to the left of $\Lambda_c$ are within the continuous spectrum). For $\beta = 1$, we observe that the MP distribution is recovered, indicating a purely uncorrelated matrix.

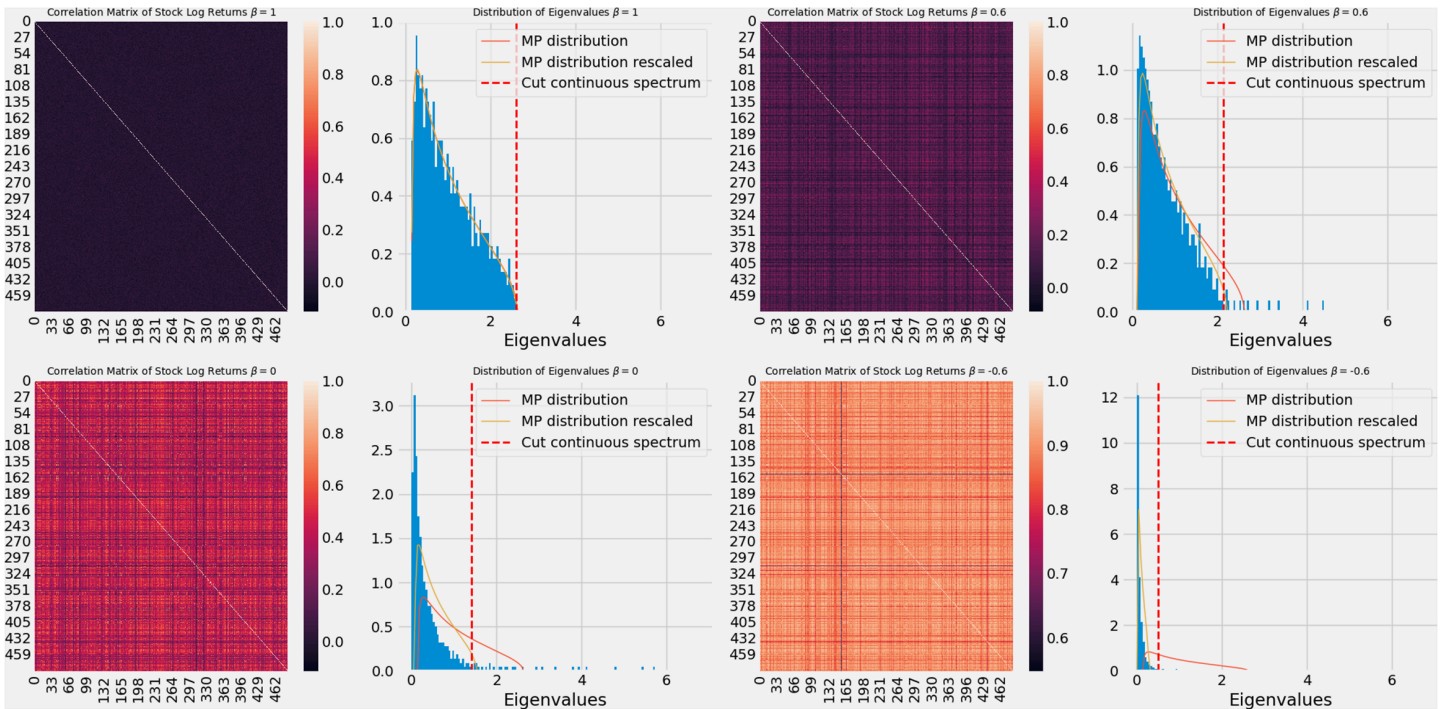

**Fig 2. Illustration of the interpolation for different values of the parameter $\beta$** On the left the entries of the correlation matrix and on the right the corresponding eigenvalue distribution with the cut.

## 4.2 Validation of the quenched average hypothesis

We begin by computing numerically $a(t)$ (Eq 12) to test whether, at low temperatures, the averaged trajectories over all eigenvalues of the continuous spectrum in Eq 10 relax to the local minimum of the potential, $a_0$. This confirms that the system exhibits self-averaging at large times, which supports the quenched average hypothesis. On the other hand, at high temperatures, we expect that trajectories are not confined to the local minimum as they do not self average. Instead they keep a momentum around the minimum of the local potential, leading to a confinement beeing proportional to the deep of the potential ($\sim 2a_0$). The left panels of Fig 3 show the behavior of $a(t)$ below the critical temperature, as calculated from Eq 28 and for $a0 = 2, 10$ corresponding to the top and lower panels. For different values of $\beta$ (signal) we compute a(t) for five different realizations of noise ($\eta_\nu$ in Eq 10). In contrast, the right panels illustrate the high-temperature regime, where the system's energy exceeds the potential well, causing trajectories to oscillate around a higher equilibrium point. Note that for negative $\beta$ of sufficiently large magnitude, the trajectory diverges in the low-temperature regime, indicating that the self-averaging assumption breaks down. In both cases, we recover the expected behavior, with variations in the local minimum potential ($a_0$).

## 4.3 Infering signal in the continious spectrum

We now turn to infering any signal form the continius spectrum by considering the correlation:

$$F_\mu(t, t_0) = \langle q_\mu(t) q_\mu(t_0) \rangle, \tag{37}$$

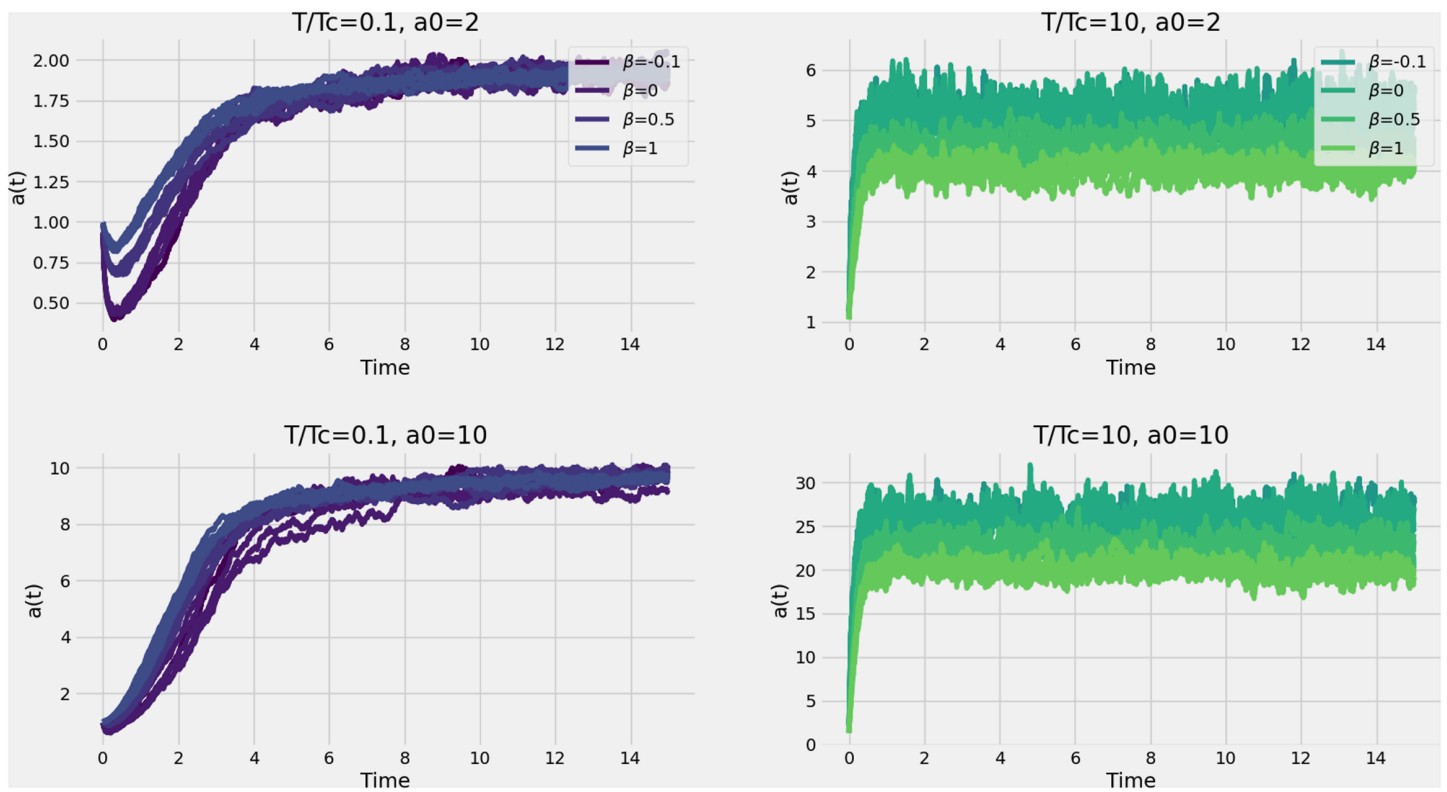

**Fig 3. Behavior of the function** $a$ **(Eq 12) for different values of** $a_0 := -h_1/h_0$ **and parameter** $\beta$, **respectively for high temperature (right panels) and low temperature (left panels).**

The brackets represent the ensemble average over the Brownian noise realizations, $\eta_\mu(t)$. We focus specifically on the case where $t_0 = 0$, and due to the chosen initial conditions, the correlation function essentially reduces to the average of the trajectories. Fig 4 shows the short-term behavior of some "eigen-trajectories", $q_\mu(t)$, for $\mu = [0, 1, 2, 50, 250]$ ($\mu = 0$ corresponds to the larger eigenvalues in the continuous spectrum of the correlation matrix) and for different degree of noise $\beta = [-0.6, -0.5, .., 0, .., 1]$, $\beta = -0.6$ corresponds to the most correlated trajectories, $\beta = 1$ corresponds to pure uncorrelated trajectories and $\beta = 0$ to the true financial stocks behaviours.

In the low-temperature regime (left panels), we observe a qualitative difference in behavior for positive versus negative values of $\beta$, as well as for small versus large eigenvalues: close to the edge at $\mu = 0$, the trajectories decay more slowly than a power law for positive values of $\beta$, and exponentially for sufficiently large negative values of $\beta$. Further from the edge (i.e., for small eigenvalues of the ECM of the returns corresponding to $\mu = [50, 250]$), the trajectories always decay more slowly than a power law. Note that at high temperatures (right figure), the system's behavior remains essentially the same as the low temperature, except for additional fluctuations.

To quantify the different regimes of signal-to-noise ratios, we show in Fig 5 the behavior of the exponents $\alpha$ and $\gamma$, defined by fitting $F_\mu(t,0) \sim e^{-\alpha t}/t^\gamma$, for different eigenvalues (labelled as $\mu$) and various values of $\beta$. The blue, red curves correspond to $T/Tc = 0.1, 10$ respectively. We observe that both the fitted values of $\alpha$ and $\gamma$ are consistent accross the different values of $\mu$ but with values that are orders of magnitude different depending on the $\mu$. This result

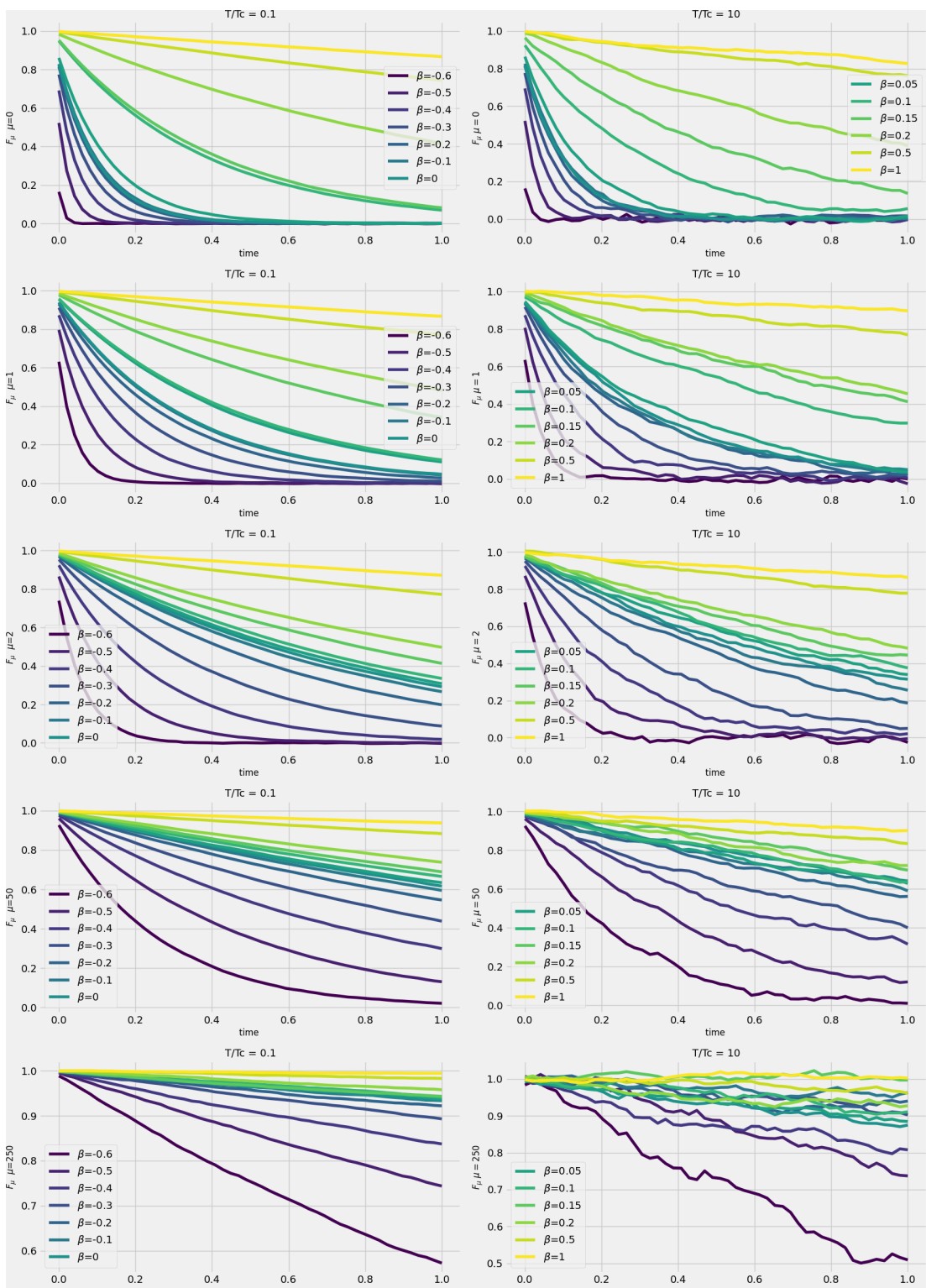

**Fig 4. Short time numerical evolution of the averaged trajectories for different values of $\beta$, different eigenvalues and different temperatures.**

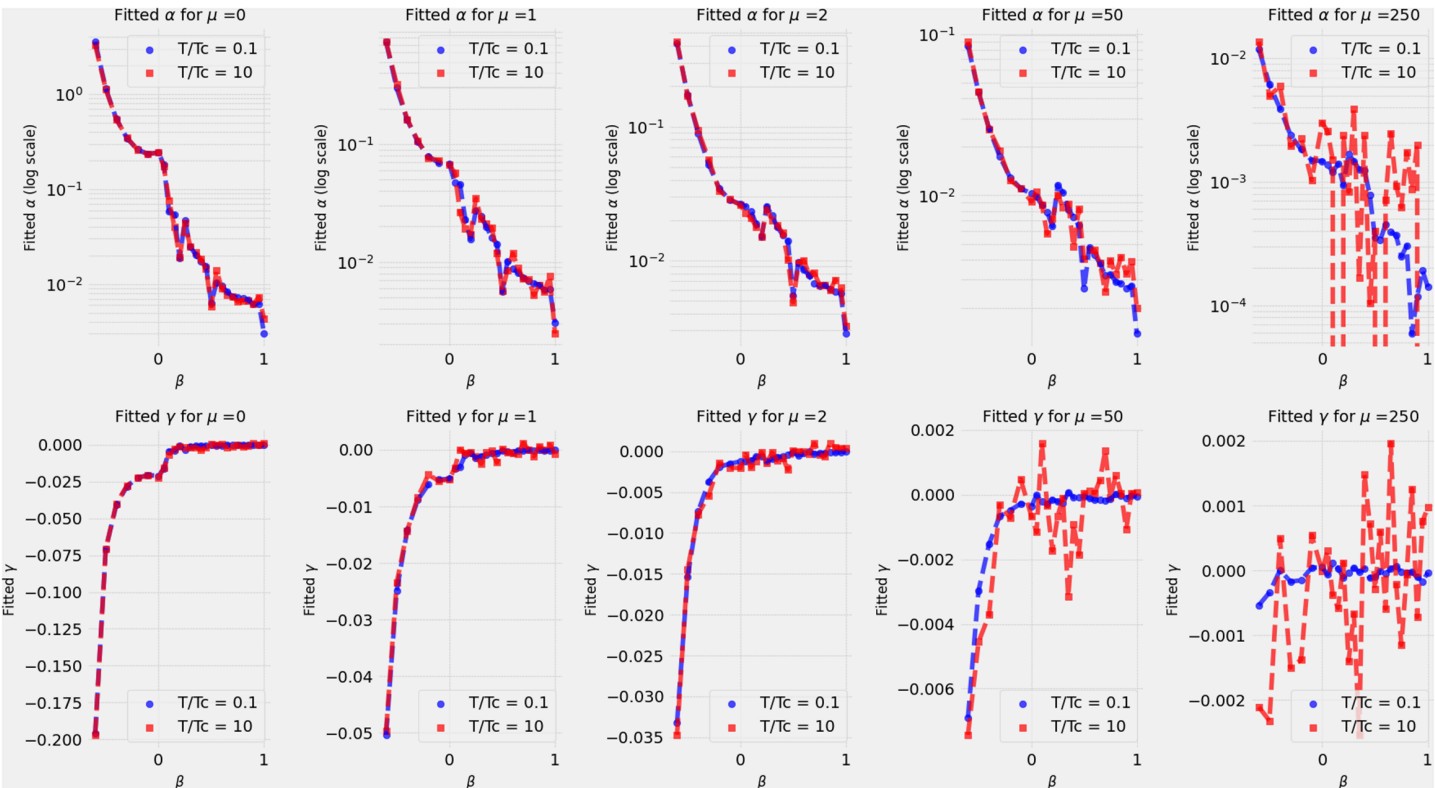

**Fig 5. Dependency on the fitting parameters α and γ on β for different eigenvalues and different temperatures.**

also holds in the high-temperature regime, though numerical instabilities are more pronounced than in the low-temperature regime. Note that the discontinuity in the function and its derivative at $\beta = 0$ is expected from the model—see equations (35) and (34).

These observations indicate that the underlying kinetics for the original data spectrum at $\beta = 0$ is qualitatively different from the kinetics of a purely Gaussian signal. These conclusions appear to contradict the observations made in [21,30] and suggest that the degrees of freedom near the cut-off ($\mu = 0, 1$) imposed by the continuity criterion from the bulk are informative.

Another indicator supporting this conclusion comes from comparing the concavity of the correlation function for different values of $\mu$ at short time intervals (the second derivative of $F_\mu$ with respect to time). In Fig 6, we show the behavior of $F_\mu$ (over 1000 realizations) for $\beta = 0$ (left panels) and $\beta = 1$ (right panels) in the low-temperature regime, where relaxation toward equilibrium is not expected for a purely random correlation matrix ($\beta = 1$).

Once again, we demonstrate that for large eigenvalues $\mu = 0, 1$, the corresponding correlation functions have a second derivative greater than 1, indicating the presence of a signal. In contrast, small eigenvalues do not exhibit this behavior.

## 5 Conclusion

In this paper, we applied random matrix theory and large $N$ methods [3] to construct and analyze formal solutions for a stochastic system governed by the correlation matrix of the financial stock returns. Our model is presented as a non-equilibrium version of a statistical inference model, capturing the correlation structure in the spectrum discussed in [9,20].

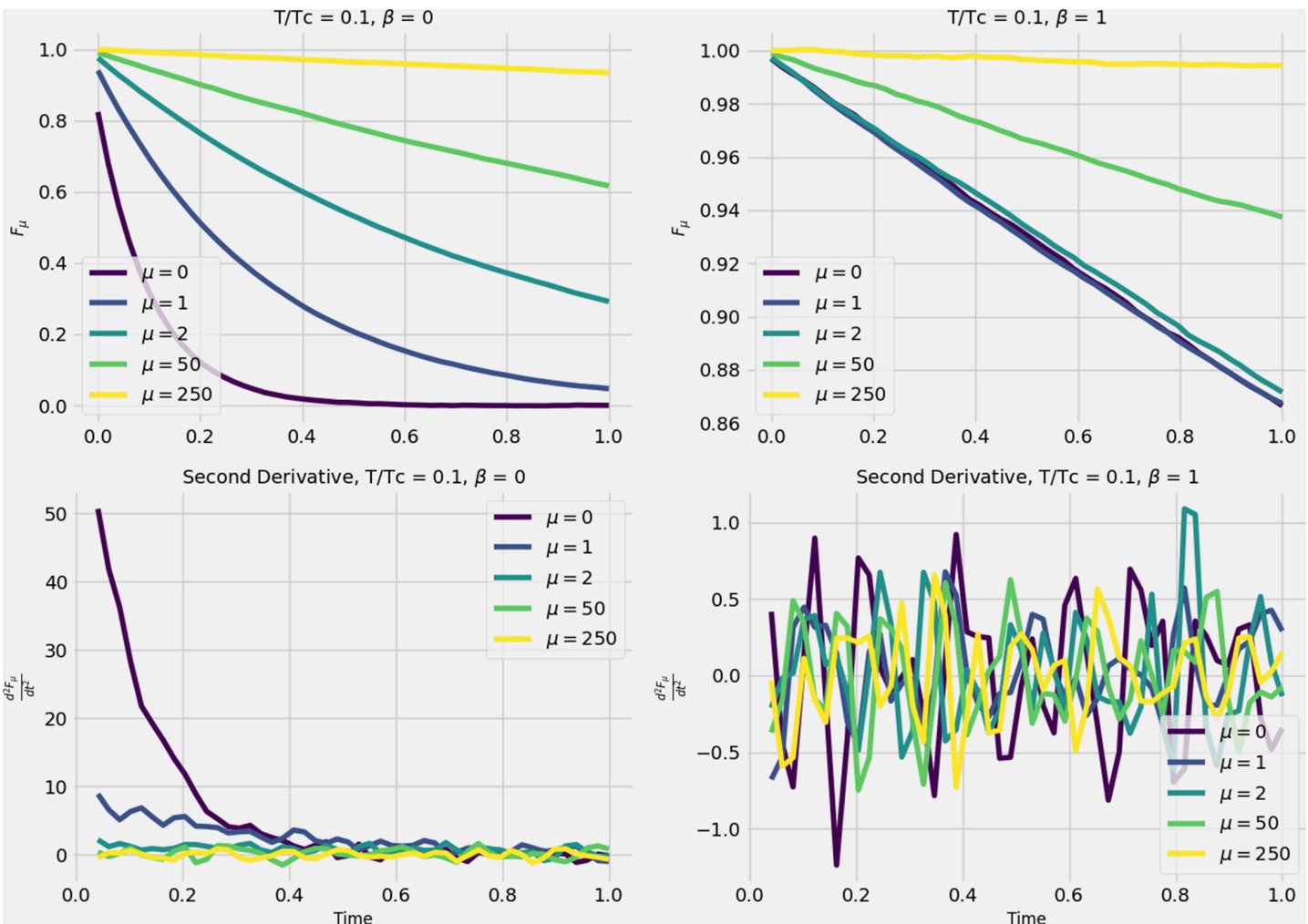

**Fig 6. On the top: Behavior of some trajectories near the larger eigenvalues for short time in the low temperature regime for the data set and a purely Gaussian matrix.** On the bottom: Behavior of the second derivative in both cases.

The underlying field theory is non-local due to the delocalized nature of the eigenvectors in a regime well described by random matrix theory.

Following [28], the equation of motion (Eq 10) can be formally solved in the quenched regime, which is justified a priori as long as the correlation matrix is well-described by random matrix theory (i.e., when $N$ is large enough and eigenvectors are delocalized). In this regime, the $O(N)$ invariant self-average, and the initial condition $q_\mu(0) = c$ for all $\mu$ is equivalent to choosing the coordinates $q_i(0)$ randomly. The solution constructed in this way, and numerically validated, reveals the existence of a critical temperature below which the system never reaches equilibrium (i.e., infinite correlation time), consistent with spin glass dynamics theory [24,26].

Our objective was to study the impact of localized degrees of freedom in the spectrum on the system's temporal behavior. To achieve this, we used real data from the S&P 500, which we corrupted with Gaussian noise via a Wiener process and interpolated between a perfectly correlated regime and a completely random one.

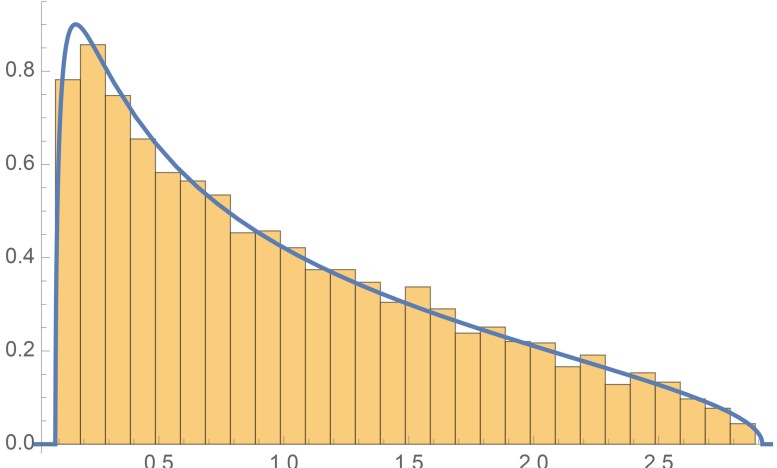

**Fig 7. Illustration of the MP theorem: Histogram corresponds to the eigenvalues for $Z$ build from a $10^4 \times 10^4$ i.i.d Gaussian matrix $X$ with variance 1.** Blue curve is the MP law for $q = 2$ and $\sigma = 1$.

Our main result shows that the evolution of the component $q_\mu(t)$ for the largest eigenvalues of the correlation spectrum exhibits different behavior than the smaller ones, depending on the level of correlation in the data. For a purely Gaussian correlation matrix, the relaxation time is very large, in agreement with the analytical predictions-see equation (14). However, as the level of correlation increases, the correlation time decreases by one or two orders of magnitude. The behavior of the component corresponding to the largest eigenvalues in the original correlation matrix at $\beta = 0$ is different from the one of a purely Gaussian matrix, pointing toward a signal detection. Indeed, we find that the underlying kinetics at the tail of the spectrum significantly diverges from what is expected for large Wishart random matrices.

## Appendix A: Marchenko-Pastur theorem

In this section we recall the standard statement in random matrix theory known as Marchenko-Pastur (MP) theorem [29]:

**Theorem.** *Let $X$ some $T \times N$ random matrix with i.i.d entries and variance $\sigma^2$. As $N, T \to \infty$ keeping the ratio $q := T/N$ fixed, the empirical eigenvalues distribution of the corresponding $N \times N$ random Wishart matrix $Z := XX^T/T$ converges weakly toward the MP distribution:*

$$\mu_{\mathrm{MP}}(x) := \frac{\sqrt{(x_+ - x)(x - x_-)}}{2\pi\sigma^2 qx}, \tag{A1}$$

where $x_\pm = \sigma^2(1 \pm \sqrt{q})^2$.

Fig 7 provides the graphical illustration of the MP theorem. Any (normalized) eigenvector of the random matrix $u^{(\lambda)}$ for some eigenvalue $\lambda$ is non-localized i.e. uniformly distributed on the $N-1$ dimensional sphere of radius 1, and the specific distribution of components $u_i^{(\lambda)}$ can be constructed as the maximum entropy distribution compatible with the constraint $\langle (u_i^{(\lambda)})^2 \rangle = 1$, the so-called *Porter-Thomas* distribution:

$$p(u) = \sqrt{\frac{N}{2\pi}} e^{-Nu^2/2}. \tag{A2}$$

## Appendix B: Inverse Laplace transform

We provide here a useful theorem about inverse Laplace transform, whose proof can be found in [31]:

**Theorem.** *Let f(t) be a locally integrable function on $[0, \infty)$ such that $f(t) \approx \sum_{m=0}^{\infty} c_m t^{r_m}$ as $t \to \infty$ where $r_m < 0$. If the Mellin transformation of this function is defined and if no $r_m = -1, -2, \cdots$ then the Laplace transformation of f(t) is*

$$\bar{f}(p) = \sum_{m=0}^{\infty} c_m \Gamma(r_m + 1) p^{-r_m - 1} + \sum_{n=0}^{\infty} Mf(n+1) \frac{(-p)^n}{n!} \tag{B1}$$

*where $Mf(z) = \int_0^{\infty} t^{z-1} f(t) dt$ is the Mellin transform of the function f(t).*

## Author contributions

**Conceptualization:** Ixandra Achitouv, Vincent Lahoche, Dine Ousmane Samary.

**Data curation:** Ixandra Achitouv.

**Formal analysis:** Vincent Lahoche, Dine Ousmane Samary.

**Investigation:** Ixandra Achitouv, Vincent Lahoche, Dine Ousmane Samary.

**Methodology:** Ixandra Achitouv, Vincent Lahoche, Dine Ousmane Samary.

**Project administration:** Dine Ousmane Samary.

**Supervision:** Vincent Lahoche, Dine Ousmane Samary.

**Validation:** Ixandra Achitouv, Vincent Lahoche, Dine Ousmane Samary.

**Visualization:** Ixandra Achitouv, Dine Ousmane Samary.

**Writing – original draft:** Ixandra Achitouv, Vincent Lahoche, Dine Ousmane Samary.

**Writing – review & editing:** Dine Ousmane Samary.

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
