## [Decision Letter · Decision Letter 0]

28 Apr 2025

PONE-D-24-46297Signal inference in financial stock return correlations through phase-ordering kinetics

in the quenched regimePLOS ONE

Dear Dr. Ousmane Samary,

Thank you for submitting your manuscript to PLOS ONE. After careful consideration, we feel that it has merit but does not fully meet PLOS ONE’s publication criteria as it currently stands. Therefore, we invite you to submit a revised version of the manuscript that addresses the points raised during the review process.

We look forward to receiving your revised manuscript.

Kind regards,

Pablo Martin Rodriguez

Academic Editor

PLOS ONE

Journal Requirements:

2. Please note that your Data Availability Statement is currently missing the repository name. If your manuscript is accepted for publication, you will be asked to provide these details on a very short timeline. We therefore suggest that you provide this information now, though we will not hold up the peer review process if you are unable.

4. Please ensure that you refer to Figure 7 in your text as, if accepted, production will need this reference to link the reader to the figure.

Additional Editor Comments:

Both reviews recommend that you revise your manuscript, so please consider making the suggested changes. The main concerns relate to the need for improvements to appeal to the journal's general audience. Having reviewed the manuscript myself, I agree that it is interesting, generally well written, and has potential for publication after major revision.

Reviewers' comments:

Reviewer's Responses to Questions

**Comments to the Author**

1. Is the manuscript technically sound, and do the data support the conclusions?

Reviewer #1: Partly

Reviewer #2: Yes

2. Has the statistical analysis been performed appropriately and rigorously? 

Reviewer #1: Yes

Reviewer #2: Yes

3. Have the authors made all data underlying the findings in their manuscript fully available?

Reviewer #1: Yes

Reviewer #2: Yes

4. Is the manuscript presented in an intelligible fashion and written in standard English?

Reviewer #1: Yes

Reviewer #2: Yes

5. Review Comments to the Author

Reviewer #1: The authors show a novel development of signal analysis, however, a more concise introduction where the tools used are bibliographically supported, as well as a review of the technique used in other research areas, is missing. It is recommended to improve the presentation of the figures throughout the text, since they are out of order with the paragraphs.

Reviewer #2: The authors propose a method to tackle the problem to understand the connected eigenvalue spectrum of the empirical correlation matrix - when the principal component eigenvalues and the bulk are not clearly separated (well described by a rescaled MP distribution), making the PCA method inefficient. The goal of this work is to set a detection threshold within the continuous part of the spectrum. They propose a method inspired in previous works using a \phi⁴ model to account Gaussian and heavy tailed distributions. The empirical correlation matrix is the 0th order approximation of the kinetic term, which is also known as Bare operators, while the corrections to this appear as a self-energy term in the bare operator, quite similar to what happens then calculating eigenvalue spectrum of wigner and Wishart random matrices using R-transform. The authors propose a decomposition of the field in the eigenbasis where the interaction terms become O(N) invariant in the large N limit. They propose a set of SDEs to model the evolution of the eigenmodes and show that the equilibrium distribution matches that one from phi⁴ model. The interesting fact is that the interactions decouple in making possible analytical calculations. They manage to obtain critical temperatures to separate different dynamical regimes. It seems that the authors want to separate the noisy region as disordered phase and true correlations as ordered phase (in context of phase transitions) so that they can separate signal from random correlations (spurious) among the data by analyzing relaxation times of eigenmodes.

The authors also provide an application in financial markets using recent data (after 2020). By carefully checking if their data obey the hypothesis of self averaging interaction term (a), they proceed in obtaining the threshold for eigenvalue spectrum in the correlation matrix and so separating signal from noise.

The work is quite interesting for mixing field theory with noise filtering, which seems to be an expanding research field (the author cites about 10 works on this topic) with fruitful insights. There are previous works from authors (such as W. Biallek) connecting the PCA method with Renormalization group theory. This work is one more advance in methods of field theory in dimensionality reduction, where the authors, by finding the relevant eigenmodes for signal filtering, are performing a renormalization in the dataset and excluding irrelevant noise. The work shows an interesting application on finance and shows that the method works well and the data seems to be good enough to put to test their theory.

I believe that the authors could expand the text in order to explain some concepts better (I had to read some references to understand and the main reference cited [9] is huge. Neither easy nor accessible). Few concepts in field theory in my opinion could be expanded and more details about the p=2 spin glass model could be added. I think the best way is to stick to a more general language instead of using field theory jargon.

The authors could also have provided possible applications with the dataset in finance. This kind of filter is useful for applications in Markowitz's modern portfolio theory and could be used for risk management. Other perspectives in different datasets such as neuron networks, MIMO and so on could be added as perspectives. The possible applications for signal prediction in machine learning and deep learning models could be good as well. The authors are free to choose the applications they have most interest in.

In my opinion the connected eigenvalue spectrum might not be exactly caused by the signal and noise being mixed in the spectrum, but also can be caused by variable variance of the data. MP distribution assumes noise with constant variance along the global time scale. Financial data such as SP500 is known for having long term memory stochastic variance (see Stanley and Mantegna Introduction to Econophysics and Voit Statistical Mechanics of financial markets). Can the authors provide explanations that the dataset can be in fact described by MP distribution? Although references provided [15, 16, 17] affirm that MP in fact was a good description of eigenvalue spectrum, this might no longer be true for 2020 data since those studies were made in 90's. Complexity of financial markets have changed a lot and it would be worthwhile if the authors could provide in fact good points that still sustents MP distribution as a good description of modern data.

6. PLOS authors have the option to publish the peer review history of their article (what does this mean?). If published, this will include your full peer review and any attached files.

Reviewer #1: No

Reviewer #2: No

---

## [Author Response · Author response to Decision Letter 1]

4 Jul 2025

Dear Editor, dear referees

Thank you very much for your message informing us of the status of our manuscript submitted under the above reference. We have carefully considered all the referees’ comments and made significant revisions, which are highlighted in blue in the new version.

Regarding the comments from Referee 2, we found it quite challenging to summarise our previous result from reference [9] given its length. In order to stay focused on the main objective of this manuscript, we have provided a concise summary and omitted some technical details that closely overlap with the content of that work.

We would like to thank all the reviewers for their thoughtful comments, which have not only enabled us to thoroughly revise the manuscript but have also greatly improved its overall quality.

On behalf of all the authors,

Best regards,

---

## [Editor Report · Decision Letter 1]

28 Sep 2025

Signal inference in financial stock return correlations through phase-ordering kinetics

in the quenched regime

PONE-D-24-46297R1

Dear Dr. Ousmane Samary,

We’re pleased to inform you that your manuscript has been judged scientifically suitable for publication and will be formally accepted for publication once it meets all outstanding technical requirements.

Kind regards,

Pablo Martin Rodriguez

Academic Editor

PLOS ONE
---

## [Editor Report · Acceptance letter]

PONE-D-24-46297R1

PLOS ONE

Dear Dr. Ousmane Samary,

I'm pleased to inform you that your manuscript has been deemed suitable for publication in PLOS ONE. Congratulations! Your manuscript is now being handed over to our production team.

Kind regards,

on behalf of

Professor Pablo Martin Rodriguez

Academic Editor

PLOS ONE